# Employment of Phenolic Compounds from Olive Vegetation Water in Broiler Chickens: Effects on Gut Microbiota and on the Shelf Life of Breast Fillets

**DOI:** 10.3390/molecules26144307

**Published:** 2021-07-16

**Authors:** Stefania Balzan, Barbara Cardazzo, Enrico Novelli, Lisa Carraro, Federico Fontana, Sarah Currò, Matteo Laghetto, Angela Trocino, Gerolamo Xiccato, Agnese Taticchi, Luca Fasolato

**Affiliations:** 1Department of Comparative Biomedicine and Food Science, University of Padova, Viale dell’Università 16, I-35020 Padova, Italy; stefania.balzan@unipd.it (S.B.); barbara.cardazzo@unipd.it (B.C.); lisa.carraro@unipd.it (L.C.); federico.fontana@unipd.it (F.F.); sarah.curro@phd.unipd.it (S.C.); matteolaghetto94@gmail.com (M.L.); angela.trocino@unipd.it (A.T.); luca.fasolato@unipd.it (L.F.); 2Department of Agronomy Food Natural Resources Animal and Environment, University of Padova, Viale dell’Università 16, I-35020 Legnaro, Padova, Italy; gerolamo.xiccato@unipd.it; 3Department of Economical and Food Science, University of Perugia, 06123 Perugia, Italy; agnese.taticchi@unipg.it

**Keywords:** olive vegetation water, phenolic compounds, broilers, gut microbiota, breast shelf life

## Abstract

Olive vegetation water (OVW) is a by-product with a noticeable environmental impact; however, its polyphenols may be reused food and feed manufacture as high-value ingredients with antioxidant/antimicrobial activities. The effect of dietary supplementation with OVW polyphenols on the gut microbiota, carcass and breast quality, shelf life, and lipid oxidation in broiler chickens has been studied. Chicks were fed diets supplemented with crude phenolic concentrate (CPC) obtained from OVW (220 and 440 mg/kg phenols equivalent) until reaching commercial size. Cloacal microbial community (rRNA16S sequencing) was monitored during the growth period. Breasts were submitted to culture-dependent and -independent microbiological analyses during their shelf-life. Composition, fatty acid concentration, and lipid oxidation of raw and cooked thawed breasts were measured. Growth performance and gut microbiota were only slightly affected by the dietary treatments, while animal age influenced the cloacal microbiota. The supplementation was found to reduce the shelf life of breasts due to the growth of spoilers. Chemical composition and lipid oxidation were not affected. The hydroxytyrosol (HT) concentration varied from 178.6 to 292.4 ug/kg in breast muscle at the beginning of the shelf-life period. The identification of HT in meat demonstrates that the absorption and metabolism of these compounds was occurring efficiently in the chickens.

## 1. Introduction

Aviculture is an efficient and sustainable animal productive system. Intensive selection plans over the last fifty years have allowed us to obtain chickens that convert feed into muscle mass with a high level of efficiency [1]. However, at the same time, the growth of the poultry industry and the spread of intensive breeding systems have led to broiler chickens being exposed to several stressors. Stressors may lead to the production of reactive oxygen species (ROS) that may injure cellular components such as lipids, proteins, and DNA, having negative consequences for growth performance and the immune response. Animals have greater exposure to disease, and an increase in lipid peroxidation reduces the meat quality [2]. The addition of antioxidants, such as phenolics, to the diets of broilers to reduce the effects of stressors and stimulate an immunity response has been proposed in several studies [3,4,5]. Phenolic compounds are natural antioxidants and antimicrobials derived from various plant materials [6]. Evidence of their beneficial effect on human health has been demonstrated by several epidemiological studies [7,8], with the health-promoting activities of dietary being confirmed by in vitro data. These studies have demonstrated an association between the consumption of phenolic compounds rich foods and a reduced risk of developing several diseases, including chronic diseases and cancer [9,10,11].

One of the peculiarities of olives and derived products is the presence of hydrophilic phenols responsible for the sensory aspects but also for biological activity. Phenolic acids and alcohols, flavonoids, lignans and secoiridoids are some of the classes of phenolic compounds presents in virgin olive oil. Secoiridoids, present in the olive drupe in the glycosidic form, are interesting for their bioactivity.

Some compounds, such as oleuropein, demethyloleuropein and verbascoside, are present throughout the fruit although often more abundant in the pulp, while nüzhenide is contained only in the seed. Oleuropein, although its concentration decreases with ripening, and demethyloleuropein are the most concentrate phenolic compounds [12].

In virgin olive oil, secoiridoids are present as final aglycon derivatives: 3,4-DHPEA-EDA from oleuropein and demethyloleuropein and, p-HPEA-EDA from ligstroside; moreover oleuropein is characterized by a bitter taste [13].

It is important to point out that the production of olive oil generates polluting by-products with antimicrobial and antioxidant activity (e.g., Olive Vegetation Water, OVW); in fact, about 90% of the phenolic compounds present in the olive are lost with the water that is released during pressing. Therefore, even if in different concentrations, in the OVW aglyconic derivatives are present similar to those of oil such as 3,4-DHPEA, *p*-HPEA, Verbascoside, 3,4-DHPE-EDA and p-HPEA-EDA [13].

Phenolic compounds can be recovered thanks to membrane technologies and could be used in the production of food and feed.

The qualitative-quantitative composition of the phenols present in the virgin olive oil and in the OVW depends on numerous factors such as the genetic and agronomic factors and, extraction-process technology [14].

The effects of the addition of olive phenolic compounds to the poultry diet have been reported in several studies with contrasting effects on growth performance [15,16,17,18,19]. The different effects observed might depend on the effective bioavailability of the different molecules. Regarding the bioavailability of phenolic compounds, the gut microbiota exerts a fundamental role, as described by several authors [20,21,22], that might operate in a reciprocal relationship. First, the gut microbiota biotransform dietary polyphenols into their metabolites, increasing their bioavailability. Second, through their antioxidant and antimicrobial activity, phenolics modulate the composition of the gut microbial community, mostly through the inhibition of pathogenic bacteria and the stimulation of beneficial bacteria. Therefore, the interactions of dietary phenolic compounds and the gut microbiota may impact host health.

In this study, the effects of the dietary supplementation of broiler chickens with OVW phenolic compounds on the gut microbiota composition, carcass yield and quality, and shelf life of breast meat were investigated. Moreover, the antioxidant activity of the phenol compound retained in the muscular tissue was studied on raw breast and cooked thawed breast.

## 2. Results and Discussion

### 2.1. Growth Performance

Table 1 reports the productive performance of broilers prior to slaughter. The groups did not show differences either during the first period (same diet) or after the experimental dietary treatment. The daily weight gain, daily feed intake, and feed conversion rate were similar among chickens fed the L0, L1, and L2 diets. No effects of dietary treatment on the slaughter yield or carcass quality were observed (Table 2). In the literature, little data on the use of olive vegetation waters in chicken feeding are available. Using different concentrations of OVW in the diet, reference [23] found greater live and carcasses weights of chickens fed phenolic compounds, but no dietary effects on the dressing percentage were reported. However, it should be noted that the live weight at slaughter of their chickens was about half that obtained in the present study. Other authors have tested the effects of extra virgin olive oil [24] or dried olive pulp inclusion [25] highlighting a positive effect of dietary integration on rearing performance. Other studies have evaluated the supplementation with olive leaves, rich in oleuropein [26], or olive leaves and olive cake [27] and have not found significant effects on rearing performance.

### 2.2. Phenol Concentration in the Diet and Breast Muscle

The concentrations of phenols in the diets used in this study are shown in Appendix A. The actual total phenol values, 175.5 and 320.2 mg/kg for diets L1 and L2, respectively, were 20.2% and 27.2% lower than the theoretical ones indicated in the paragraph 2.2. The concentration of hydroxytyrosol (HT) was 97 and 174.6 mg/kg for L1 and L2, respectively. In the breast muscle, among the phenol compounds in the diet (Appendix A), only HT was found (Table 3). Its value at the beginning of the shelf-life period (24 h) was proportional to the concentration of HT in the feed. The absorption of tyrosol (T) and HT in humans is dose dependent [28], while in poultry, it seems that no more than 10% of phenols are absorbed in the small intestine [29]. After 10 days of storage at 4 °C, the residual concentration of HT in breast muscle was no longer detectable in the L1 group, while it was approximately one-fourth of the initial concentration in the L2 group. In a sample of the control group, the measured HT concentration was 78 µg/kg. According to de la Torre [30], the presence of HT in the biological fluids of human volunteers, even after several hours of fasting, is the result of dopamine metabolism. Studies conducted on animal models have shown that especially in the presence of alcohol, the metabolism of dopamine produces significant quantities of HT in addition to the usual terminal product represented by 3,4-dihydroxyphenilacetic acid. Branciari et al. [31] assessed chicken diets with a total phenolic content (mainly T and HT) ranging from 14 to 24 mg/kg (approximately 12 times lower than those used in the present study). The authors identified T, rather than HT, in samples of breast muscle, in concentrations ranging from 8 to 47 µg/kg. This was 5 to 33 times lower than the concentrations measured in the breast samples of the L2 group of the present study. They found only sulphate metabolites of HT in minor concentrations in the muscular tissue.

### 2.3. Gut Microbiota Profiles

The gut microbiota composition was determined from cloacal swabs collected from broilers during the breeding period at 23, 34, and 44 days of age. The DNA was extracted and used as a template for rRNA16S amplicon library preparation. Libraries were sequenced using Illumina technology (Illumina^®^, San Diego, CA, USA). The composition of the gut bacterial community was analyzed in order to evaluate the effects of OVW polyphenol dietary supplementation. The multivariate analysis demonstrated a significant effect of age of animals on the gut bacterial community composition (*p* < 0.001, R^2^ = 0.261). On the contrary, the diets enriched with CPC seemed not affect the gut microbiota. The substantial alterations of the microbiota related to the age of the animals have been widely reported in the literature [30,31,32,33]. At 23 days of age, the microbial community was dominated by Proteobacteria (mostly by members of the Enterobacteriaceae family); however, the concentration of these species reduced during the growth of the animals with concomitant increases in the Firmicutes and Bacteroidetes phyla (see Figure 1A,B). The Lactobacillaceae family and the Ruminococcaceae and Lachnospiraceae families (both belonging to Clostridia class) increased in concentration at 34 and 44 days of age, respectively (Figure 1B). The biodiversity (reported as the Shannon alpha-diversity index) increased from 23 to 34–44 days of age (Figure 1C). The expected modification of the microbiota composition linked to the administration of different diets was difficult to identify due to the wide intra-individual variability (related to the shorter gut tract and faster intestinal transit in poultry than in other food animals) and the consistent changes related to age [34,35].

### 2.4. Sensory Shelf Life, pH, and Drip Loss

Figure 2A reports the SI evolution during storage. Breasts maintained their freshness scores until 200 h of refrigeration. After this period, rapid sensory decay was observed. Samples from diets L1 and L2 had sensory shelf lives of 256–263 h, while most breasts from animals that consumed the control diet did not overcome the demerit threshold (SI = 1.8) of 264 h. Samples with SI lower than 1.8 were considered spoiled.

Statistical analyses revealed that supplementation with CPC only affected the sensory index at 156 and 256 h (*p* < 0.05). SI suggested that the shelf-life behavior of samples differed, especially in the latest phase of conservation.

Dietary treatment did not affect the pH (*p* > 0.05; Figure 2C) which, instead, was affected by the storage time (*p* < 0.001; Appendix A). For the drip loss (Figure 2B), in addition to time (*p* < 0.001; Appendix A), the effect of dietary treatment was also statistically significant (*p* < 0.001) with L1 having the lowest value (Appendix A). Branciari et al. [31] reported that the pH after 24 h and drip loss of chicken meat were not affected by the diet integration with polyphenols derived from semi-solid destoned olive cake. Changes in the pH and drip loss during the storage period are common findings and are also affected by the type of packaging [36].

### 2.5. Shelf-Life of Breast Fillets Determined with Culture-Dependent and Culture-Independent Methods

The effect of CPC supplementation seemed to affect some microbial targets (TVC; *Pseudomonas, Shewanella* and *Enterobacteriaceae*) but only after 216 h of conservation (Appendix A). Confirming the culture-dependent results, a comparison of the microbial community profiles assessed by rRNA16S amplicon sequencing demonstrated a strong effect of time (*p* < 0.001). During the shelf-life period, the biodiversity gradually reduced and the composition of the community was clearly modified (see Appendix A), with an increase in the relative abundance of *Proteobacteria* and decreases in *Firmicutes* and *Bacteroidetes* (Appendix A). At 11 days (264 h of conservation), as for the cultural methods (see below), differentiation of the community depending on the presence of CPC was evident. The Principal Coordinate Analysis (PCoA), presented in Figure 3B, demonstrated this effect, and this was confirmed by the Adonis multivariate analysis (T11 *p* < 0.001). In terms of the composition of the community, the culture-independent analysis (resumed in the heat map presented in Figure 3A) confirmed the presence of meat spoilers such as *Pseudomonas*, *Acinetobacter,* and *Shewanella,* that progressively increased their presence in the community during the shelf-life period, while the concentration of bacteria from fecal contamination (*Fecalibacterium*, *Coprococcus* and *Escherichia*) reduced.

In general, L1 and L2 diets were associated with higher levels of *Pseudomonas* spp., a specific group of spoilage organisms that are mainly involved in the sensory decay of poultry meat [37].

These results agree with previously reported data on normal breasts under refrigeration [38] with a slightly lower microbial load in the latest part of the shelf-life period.

Few works have reported the effects of phenols resulting from waste products of the olive oil industry in the broiler diet [29]. Therefore, the effects of olive mill wastewater extract on the microbial quality and shelf life of chicken breasts have rarely been investigated. The reported results suggest a small effect of the phenols retained inside the breast meat, with an increase in microbial growth of spoiler targets during the last period of conservation. As previously reported in in vitro studies, similar extracts show antimicrobial activities, especially in gram-positive bacteria [39]. Moreover, supplementation with olive mill wastewater concentrate seems to reduce the incidence of foodborne pathogens, such as *Campylobacter* spp., in the feces of broilers [23]. Here, the retained phenols inside the breast could modify the microbial environment, favoring the presence of *Shewanella* and *Pseudomonas*. However, mild impacts on sensory data during the shelf-life period were observed.

The growth parameters of TVC and *Pseudomonas* are reported in Table 4. The microbial growth was very similar among diets with an initial Lag phase for TVC during the first 5–7 days of refrigeration. Due to their psychrotrophic habits, *Pseudomonas* spp. showed a very limited adaptation period (Lag time 0–4 days). Microbial growth, as shown by the plateauing of parameters, overcame the specific spoilage thresholds suggested for chicken breast samples stored in air [37]. Growth parameters allowed the estimation of microbial shelf life, which ended earlier than sensory decay in all observed cases. When the microbial load exceeded 7 log_10_ CFU/g, particularly for *Pseudomonas* spp., many spoilage mechanisms, including amino acid degradation, slime, and off-odor formation, increased, and there was a faster decrease of sensory attributes. As observed for the SI, the estimated microbial shelf life was longer in the sample that consumed the control diet. Considering *Pseudomonas* as a specific spoilage marker, it is possible to assume 9.5-day shelf life for L1 and L2 breasts as compared with 11 days for the control samples (Table 4).

### 2.6. Proximate Composition, Cooking Loss, and Fatty Acid Composition

The composition analysis did not show differences between the samples of the control group and those with diet added of CPC (Table 5). The cooking treatment caused significant increases in protein and fat, which were dependent on the overall increase in dry matter following the loss of water during cooking. For the dry matter values, only a significant difference for the ash concentration was shown. No differences were observed with respect to cooking loss. Branciari et al. [31] did not find any differences in the proximate composition of thawed breasts from Ross 308 broilers fed diets supplemented with a semi-solid olive cake at 82.5 g/Kg and 165.0 g/Kg of diet. Giannenas et al. [40] feeding Ross 308 up to 42 days with experimental diets supplemented with oregano blend (300 g/ton) plus attapulgite (3 kg/ton) and with a mix of oregano and laurel (500 g/ton) did not find differences on proximate composition neither on breast nor on thigh meat. Starčević et al. [41] supplementing the feeds for chicken Ross 308 with thymol (200 mg/kg), tannic acid (5 g/kg) and gallic acid (5 g/kg) up to 35 days found higher fat content and lower protein content in breast meat of the tannic acid group compared to control one. At the same time the feed intake of the experimental groups was higher than control group, suggesting that the resulting extra energy produced could have been deposited as fat in the muscular tissue. The addition of fermented or enzymatically fermented dried olive pomace to broiler chickens’ diet (Ross 308) at three inclusion levels (7.5, 15 and 30%) significantly increased protein content and decreased fat in breast muscle of the experimental groups [42]. Table 6 presents the fatty acid profiles of the breast muscle. No significant variation was shown among experimental groups, except for significantly higher percentages of the two essential fatty acids (C18:2 n6 and C18:3 n3) in chickens that consumed the L2 diet. Although without statistical significance, the concentration of long-chain polyunsaturated fatty acids showed an opposite trend, with values tending to be higher in the control group. The relative abundance of essential fatty acids (18:2 n6 and 18:3 n3) and their long-chain polyunsaturated derivatives (20:4 n6, 20:5 n3, 22:5 n3, 22:6 n3) depends also by the activity of tissue enzymes with desaturase function. Therefore, the regulation of enzymatic activity can cause a consistent variation in the concentration of these fatty acids in the liver and other peripheral tissues. The phenols, and among these hydroxytyrosol, can under certain conditions affect the lipid metabolism also through the regulation of desaturase activity. Valenzuela et al. [43] observed that dietary supplementation with 5 mg/day of hydroxytyrosol in mice did not cause changes in either the blood or tissue lipid profile. Administration of a high-calorie diet causes an immediate decrease in the hepatic concentration of total n6-LCPUFA and total n3-LCPUFA in mice. This effect is rebalanced when the high-calorie diet is added with 5 mg of hydroxytyrosol. HT therefore seems to have a normalizing effect on the desaturase activity (mainly Δ-5 and Δ-6 desaturase) in conditions of food stress. On the contrary, sesamin (a phenol ascribed to lignans group) has shown an inhibiting action on rat liver Δ-5 desaturase [44]. Therefore, in the experimental conditions of the present study in the absence of both food and environmental stress factors, the integration of the diet with an extract rich in polyphenols does not seem to determine significant changes in the enzymatic desaturase activity and consequently the lipid profile of the breast muscle tissue was unaffected.

### 2.7. Lipid Oxidation

Lipid oxidation was studied through the measure of the secondary (TBARs) and primary (D232 and D270) oxidation compounds (Table 7). No significant differences were noted among the three experimental groups, although it is worth noting that there was a slight increase in the TBARs values for chickens in the L2 group with respect to those in the L1 and L0 groups, while D232 showed the opposite trend. These two analytical indicators were significantly correlated (TBARs vs. D232 r = −0.5, *p* < 0.001 and TBARs vs. D270 r = −0.37, *p* < 0.05). The cooking treatment caused the decreases in the primary oxidation products (*p* < 0.01), which were followed by increases in the secondary ones (*p* < 0.001). No interactions between diet and cooking were observed. On the contrary, [31] and [45] observed a significant reduction in TBARs in the breasts from chickens whose diets were supplemented with either Olive cake or OMWW. Similarly, [46] observed a reduction in TBARs in the quadriceps femoris of chickens whose diets were enriched with OMWW permeate and retentate. Oke et al. [16] added different volumes of olive leaf extract (5, 10 and 15 mL/liter) to the drinking water supplied to Abor acre strain broilers, observing a significant effect on the reduction of blood levels of malondialdehyde (MDA). Ibrahim et al. [42], observed a significant reduction in the MDA concentration in chicken breast obtained by adding the diet with increasing levels of fermented or enzymatically fermented dried olive pomace. Unlike other studies, a protective effect against muscular fat oxidation in muscle tissue was not observed in the present one. There are many factors that generate variability and differences between studies. Apart from the animal breed and the length of the fattening period, other factors such as the nutrients composition of the diet, the feed intake, the environmental temperature together with other stressful conditions can lead to very different results that often are difficult to compare. Not last the type of matrix with which the diet is enriched with phenols (olive-cake, olive pomace, olive leaf extract, vegetative water extract, just to name a few) due to the concentration of the active compounds and the presence of additional substances with a synergistic or protective action present in the same matrix, can significantly influence the antioxidant action both in vivo and in meat tissue after slaughter [47]. The potential health benefits from integrating antioxidant additives in fresh meat and meat products are not always proven [48]. On the contrary, many primary and secondary lipid and protein oxidation compounds, such as hydroperoxides, epoxides, 4-hydroxynonenal, malonaldehyde are recognized as potential carcinogens or can affect cellular signal transduction as is the case of carbonyl compounds [49]. In the case of fresh meat, the possibility of increasing the antioxidant potential through the rationing of animals is a very suggestive prospect, especially in the use of natural rather than synthetic substances. In fact, unlike meat-based products where technology offers different possibilities of intervention to increase the antioxidant potential, in fresh meat the only alternative to intervention through the diet of animals is that of a treatment aimed at the surface of the product that however it must be consistent with the legislation in force. The residual quantity of phenols measured in the breast meat of the present study is unlikely to cause direct health benefits, but indirectly can do. The cooking of meat, for instance, causes a net loss of the tissue antioxidant defenses, therefore the meat tissue enrichment by compounds that are still active even after heat treatment, as is the case of phenols, is of great application interest in view of an increase in the shelf life and safety of the product itself. The reason why neither in vivo nor ex vivo effects were observed as a consequence of the feeding trial with CPC, deserves further experimental investigations, especially regarding the interaction between dietary polyphenols and the intestinal microbiota of the chicken on which it leverages a significant part of the potential biochemical effects of phenols [9,10].

## 3. Materials and Methods

### 3.1. Experimental Facilities

The trial was undertaken in the poultry house of the Experimental Farm of the University of Padova (Legnaro, Padova, Italy) after 6 months of downtime. The poultry house was equipped with a cooling system, forced ventilation, radiant heating, and controlled light systems. A total of 12wire-net pens (2.5 × 2.4 m; 6 m^2^) were used, each equipped with five nipple drinkers (distance: 20 cm) and a circular feeder (diameter: 30 cm) for manual distribution of feed. Each pen had a concrete floor covered with wood shaving litter (depth 5 cm, 2.5 kg/m^2^).

Light was provided for 24 h/day for the first 2 days after the chickens had arrived at the poultry house. Then, the number of hours of light was progressively reduced until an 18L:6D photoperiod was achieved, which was then maintained from 12 days of age onwards.

### 3.2. Animals, Experimental Groups, and In Vivo Recordings

The study was approved by the Ethical Committee for Animal Experimentation (Organismo per la Protezione del Benessere Animale; OPBA) of the University of Padova (project 17/2016; No 154392 of the 10 May 2016). All animals were handled according to the principles of Directive 2010/63/EU [50] regarding the protection of animals used for experimental and other scientific purposes. Researchers involved in animal handling were either animal specialists (PhD or Master’s degree in Animal Science) and/or veterinary practitioners.

A total of 144 1-day-old chicks (Ross 308) were delivered by a commercial truck, in compliance with Council Regulation (EC) no. 1/2005 [51], to the experimental facilities of the University. All chicks had been vaccinated against Marek’s disease, infectious bronchitis, and Newcastle disease at the hatchery. Chicks were individually weighed on the day of their arrival, identified by a leg mark, randomly allocated among the 12 pens (12 birds per pen and 10 birds/m^2^), and then weighed once per week to measure their live weight until commercial slaughtering. Pen feed consumption was measured daily during the trial. Three commercial diets were administered in crumble form during the trial (Martini SpA, Longiano, Forlì-Cesena, Italy): diet P1 from 0 to 23 d, diet P2 from 24 to 37 d, and diet P3 from 38 d until slaughtering at 48 days (see Supplementary materials for analytical data, Appendix A). The crude phenolic concentrate (CPC) was obtained as described by [12], briefly OVWs were treated using enzymes with pectinase and hemicellulosic activities and then subjected to microfiltration, ultrafiltration and reverse osmosis. CPC was added to the diets as follows: (i) L0 Control diet (P3), (ii) L1 Control diet plus CPC (220 mg/kg feed as theoretical total phenols), (iii) L2 Control diet plus CPC (440 mg/kg feed as theoretical total phenols). The actual content of individual phenolic compounds of CPC and of the diets were reported in Appendix A. Each diet was replicated in four pens (homogeneous for initial live weight and variability). The CPC was supplied from 24 days (d) of age until commercial slaughter at 48 d (see Supplementary Material for phenolic composition of CPC). All animals were fed ad libitum throughout the trial, and the supplemented diets were prepared weekly.

To determine the composition of the gut microbial community, cloacal swabs were collected (4 animals/pen) at 23, 34, and 44 d of age for a total of 12 animals (4 on each diet) who were tested three times (36 swab samples collected in total).

### 3.3. Commercial Slaughtering, and Carcass and Meat Quality Recording

At 48 days of age, after feed and water withdrawal (for 7 and 2 h, respectively), all chickens were slaughtered in a commercial slaughterhouse. Chickens were individually weighed before crating. All chickens from a pen were loaded into a transport cage (height, 62.5 cm × 160 cm × 25.0 cm; floor area, 1 m^2^). Loading and transport from the experimental facilities to the commercial slaughterhouse and lairage before slaughter took approximately 3 h. Chickens were slaughtered according to the standard practices of the commercial slaughterhouse. Carcasses were recovered after 2 h of refrigeration at 2 °C and individually weighed to measure the slaughter dressing percentage [52].

Twenty-four hours after slaughter, carcasses were dissected to obtain the pectoralis major muscles separated from the breasts [53].

During the trial, there was a recorded loss of 17.4% (25 chickens) due to mortality. Of the 119 carcasses, 72 were assigned to microbiological analyses during the shelf-life period (Section 2.4 explains the establishment of the breast microbiota composition by culture-dependent and -independent methods), while the remaining 47 were frozen (−40 °C) and subsequently used for the evaluation of the proximate compositions and lipid oxidation rates of raw and cooked meat (paragraph 3.5).

### 3.4. Shelf-life Evaluation

#### 3.4.1. Preparation of Packaged Chicken Breast

*P. major* muscles were individually packaged in low-density polyethylene trays wrapped in a 12 µm-thick PVC film (Weegal, KOEX 412, Gruppo Fabbri, Vignola, Modena, Italy) and stored at 4 ± 1 °C. in a refrigerated cabinet (Majolo^®^ Plus 100 Seasoning Controller, Majolo, Cadoneghe, Padova, Italy). Exposure to light was fixed from 8:00 to 20:00 using a 36 W fluorescent lamp [54]. The storage settings were designed with the intent to mimic the refrigeration conditions during the sale. The packages were randomly sampled at 24, 72, 120, 180, 216, and 264 h from the slaughter date

#### 3.4.2. Sensory Analysis

A sensory evaluation of fresh breasts was performed according to [37] using a demerit 3-point scoring system (1 = not acceptable, 2 = acceptable, 3 = good quality). The synthetic sensory index (SI) was calculated as [(2 × odour score + 2 × colour score + 1 × texture score)/5], with a score of 1.8 acting as the threshold to define spoiled samples. The sensory analysis was performed by eight trained panellists.

#### 3.4.3. Microbiological Analysis of Breast Meat during the Shelf-Life Period

Microbiological analyses were performed according to the procedures described by [38]. Pectoralis major muscles (about 2 cm long × 1 cm deep × 2 cm wide) were aseptically excised along each breast to obtain a representative sample of 25 g per breast. Samples were mixed in a stomacher bag with 225 mL of buffered peptone water and analyzed after the appropriate dilutions. For DNA extraction, 1 mL of each homogenate was collected in 2 mL Eppendorf tubes and centrifugated at 13,500× *g* for 1 min (Eppendorf centrifuge 5425, Hamburg, Germany). Then, the supernatant was discarded and the pellet was frozen at −80 °C. Several microbial targets were investigated to describe the dynamics of the microbial population during the shelf-life period as follows: The total viable count (TVC) and total psychrotropic count (TPC) were evaluated on Plate Count Agar (Biokar Diagnostics, Beauvais, France), and plates were incubated at 30 °C for 72 h or at 4 °C for 10 days. Enterobacteriaceae were enumerated with Violet Red Bile Glucose Agar (Biokar Diagnostics, Beauvais, France) at 37 °C for 24 h. Lactic acid bacteria (LAB) were analyzed on De Man, Rogosa, and Sharpe Agar (Biokar Diagnostics, Beauvais, France) under anaerobic conditions at 30 °C for 48 h. The *Pseudomonas* spp. count was evaluated on *Pseudomonas* Agar Base supplemented with cetrimide, fucidine, and cephaloridine at 25 °C for 48 h (Oxoid Ltd., Basingstoke, Hampshire, UK). The H_2_S-producing bacteria (putative *Shewanella* spp.) count was carried out on iron agar (Lyngby, Laboratorios Conda, Torrej’on de Ardoz, Spain) at 25 °C for 48 h. Results are reported as log_10_ CFU/g meat.

#### 3.4.4. pH and Drip Loss

A pH meter (Portamess^®^ 910, Knick, Berlin, Germany), equipped with a specific electrode (Mettler Toledo, Milano, Italy), was used to measure pH values by insertion in the pectoralis major muscles in triplicate on their ventral side. Drip loss was evaluated in accordance with the method presented by [55] using the bag method. Approximately 2.5 cm slices cut from the surface of dorsal breasts were inserted into polyethylene bags and suspended overnight at 2 ± 1 °C. The drip loss (%) was calculated by the following equation: [(initial weight − final weight)/initial weight].

#### 3.4.5. Phenol Concentration in the Diet and in the Breast Muscle

The extraction of phenols was conducted on 5 g of minced diet sample mixed with 50 mL of methanol/water (80/20 (*v*/*v*) solution containing 20 mg/L of butylated hydroxytoluene (BHT), the mixture was homogenized using a rod disperser (IKA, T50 Ultra-Turrax, Werke, Staufen, Germany) for 1 min at 7000 rpm, centrifuged at 9327× *g* for 10 min and the supernatant recovered. The procedure was repeated twice, and the collected extract was then concentrated by a rotary evaporator (Buchi Rotavapor, R-210, Switzerland) until reaching a final volume of 20 mL. A SPE Bond Elut Jr-C18, 1 g cartridges (Agilent Technologies, Santa Clara, CA, USA), previously activated with 10 mL of methanol and 10 mL of water, was loaded with 1 mL of aqueous extract the elution was performed with 50 mL of methanol. After solvent removal under vacuum, the phenolic extract was solubilized in 1 mL of a solution composed of methanol/water (50:50 *v*/*v*) and filtered through a polyvinylidene fluoride (PVDF) syringe filter (0.2 mm). The qualitative and quantitative analysis of the phenolic compounds of the extract was conducted according to [31] by HPLC (Mod. 1100 Agilent Technologies, Santa Clara, CA, USA), equipped with a C18 column (Spherisorb ODS-1 (250 mm × 4.6 mm) 5 μm particle size, supplied by Waters S.p.A. (Milan, Italy). The phenolic compounds were detected by using the DAD set at 280 nm. The quantification of polyphenols was determined by using single calibration curves for each compounds and the results are expressed as mg kg^−1^. The hydroxytyrosol (3,4-DHPEA) was obtained from Cabru S.p.A. (Milan, Italy), tyrosol *p*-HPEA was purchased from Merck Life Science S.r.l. (Milan, Italy), and verbascoside was taken from Extrasynthese (Genay, France). The dialdehydic form of the decarboxymethyloleuropein aglycone (3,4-DHPEA-EDA) was extracted from virgin olive oil according to the procedure reported from [56].

A total of 10 grams of meat were mixed with 50 mL of methanol and water (80/20, *v*/*v*) containing formic acid 0.2% and 20 mg/L of BHT. The solution was homogenized using a rod disperser (IKA, T50 Ultra-Turrax, Werke, Staufen, Germany) for 1 min at 7000 rpm, centrifuged at 4146× *g* for 10 min and the supernatant recovered. The operation was repeated twice, and the two aliquots of the extract were collected and the methanol was removed by rotary evaporator (Buchi Rotavapor, R-210, Switzerland). Twenty mL of the aqueous extract was loaded to SPE Bond Elut HF Mega BE-C18, 5 g cartridges (Agilent Technologies, Santa Clara, CA, USA), previously activated with 20 mL of methanol and 20 mL of water; the elution was performed with 50 mL of methanol. After solvent removal under vacuum, the phenolic extract was recovered with 0.5 mL of a solution composed of methanol/water (50:50 *v*/*v*) and filtered through a polyvinylidene fluoride (PVDF) syringe filter (0.2 mm). The HPLC analyses of the phenolic extracts were conducted according to [56] with the same equipment as reported above. The hydroxytyrosol was detected by using the fluorescence detector (FLD) operated at an excitation wavelength of 280 nm and an emission of 313 nm. The quantification of hydroxytyrosol was made using the calibration curve and the results are expressed as µg/kg.

### 3.5. Proximate Composition, Cooking Loss, and Fatty Acid Analysis

The proximate composition, fatty acid profile, and oxidative stability were evaluated on both raw (right side P. major) and cooked (left side P. major) samples. After thawing, the cooking loss was evaluated according to [53] with the following modifications: The left half of each breast was weighed, vacuum packed, and cooked in a water bath at 80 °C for 45 min. After that, the breasts were blast chilled up to 4 °C and kept in the refrigerator (4 °C) for 72 h before being unpacked and weighed again to calculate the cooking loss as [(raw meat weight − cooked meat weight)/raw meat weight] × 100). The muscle, both raw and cooked, was finely ground (twice 2500 rpm × 10 s, Retsch, Dusseldorf, Germany) and submitted to moisture, crude protein, and ash [57] analysis, while the crude fat content was determined using the method presented by [58]. Briefly, 200 mL of dichloromethane/methanol 1:1 mixture was added to 5 g of the sample, homogenized at 11,000/min for 1 min (Ultraturrax T25 Basic, Ika Werke. Staufen, Germany), and incubated at 60 °C in an oven (PID system, M120-VF, MPN instruments, Bernaggio, MB, Italy) for 20 min. Subsequently, another 100 mL of dichloromethane was added, and after homogenization (11,000/min for 1 min) the sample was filtered (rapid filter paper) and 100 mL of KCl 1 M were added to the permeate. After centrifugation (Avanti J-E, Beckman Coulter, Brea, CA, USA) at 1000× *g* for 30 m at 4 °C, the supernatant was discharged and the organic extract was filtered through anhydrous sodium sulphate. An aliquot (around 10 g) was used for gravimetric determination of the fat percentage, the remainder was distilled (Rotavapor^®^ R-210, Büchi, Essen, Germany), and the anhydrous fat was used for further analysis. Fatty acid methyl esters (FAMEs) were prepared by weighing 0.015 g of anhydrous fat into a glass tube with a screw cap [39], which was solubilized with 1 mL of 3 M methanolic HCl (Sigma-Aldrich, St. Louis, MO, USA) and incubated for 2 h at 60 °C. Then, 1 mL of deionized water was added and, after mixing, the FAMEs were recovered in 1 mL of n-hexane. FAMEs (1 μL) were analyzed by a gas chromatograph (Shimadzu Italia Srl, model 2014, Milano, Italy) equipped with a flame ionisation detector (set at 250 °C) and a split-splitless injector (set at 280 °C). The thermal chamber, which held the capillary column (Supelco SP-2560; 100 m × 0.25 mm × 0.2 μm), was kept at 60 °C for 8 min. The temperature was then increased to 120 °C for 1 min at a rate of 10 °C/min, and then from 120 to 240 °C at a rate of 2.5 °C/min. The final isotherm was kept at 240 °C for 20 min. Fatty acids were identified using an external standard mixture (37 Component FAME Mix, Supelco, Germany). Analyses were done in duplicate.

### 3.6. Lipid Oxidation

Primary oxidation compounds were determined as conjugated dienes by spectrophotometric analysis in the ultraviolet field using a modified version of the method indicated in annex III of the Council Reg. EU/2015/1833 [59]. Briefly, 0.01 ± 0.001 g of anhydrous fat was weighed into a 10 mL graduated flask and dissolved by spectrophotometric grade cyclohexane up to the mark. An UV/Vis spectrophotometer (Mod. 7800 Jasco UV/VIS, Oklahoma City, OK, USA) was used to determine the specific extinction values at 232 (conjugated dienes) and 270 nm (conjugated trienes) using the same solvent as the reference. Secondary products of lipid oxidation were determined according to the method presented by [60] with modifications, as follows: in a centrifuge tube, 8 mL of an aqueous solution of 5% (m/v) trichloroacetic acid and 5 mL of n-hexane were added to approximately 2 g of the homogenized sample. After homogenisation with Ultraturrax for 30s at high speed, the sample was centrifuged for 3 min at 3000× *g* at 4 °C (Eppendorf 5810R; Eppendorf, Hamburg, Germany), and the supernatant was removed. The sample was filtered through a rapid paper filter, and 2.5 mL of the filtrate was added to 2.5 mL of an aqueous solution of 0.02 M thiobarbituric acid and incubated for 35 min at 95 °C in a thermostated bath (Julabo ED-13; Seelbach, Baden-Württemberg, Germany). After cooling, the absorbance was taken using a spectrophotometer set at a wavelength of 532 nm. Data are expressed as milligrams of malondialdeyde per kilogram of meat. The measurements were carried out in duplicate on both cooked and raw chicken breasts.

### 3.7. DNA Extraction and NGS Library Preparation

DNA was extracted using a DNeasy PowerSoil^®^ DNA Isolation Kit (Qiagen, Hilden, Germany). The homogenate pellets of breast meat were defrosted with 600 uL of Lysis buffer provided by the kit. Cloacal swabs were cut with sterile scissors, put into microtube containing 600 uL of Lysis buffer and 6 uL of 2-mercaptoethanol (Sigma-Aldrich, St. Louis, MO, USA). Tubes were mixed well by using vortex for 5 min and swabs were removed. For both cloacal samples and breast meat pellet a total of 100 mg of sterile 0.1 mm beads (BioSpec, Bartlesville, OK, USA) was added, and a bead mill (TissueLyser, Qiagen, Hilden, Germany) was used to carry out disruption in high-speed shaking steps (2 × 30 s at 30 Hz). A total of 40 uL of proteinase K (Qiagen, Hilden, Germany) was added and incubated at 56 °C for 90 min. After this step, the DNeasy PowerSoil^®^ DNA Isolation Kit manufacturer’s instructions were followed. The concentration and purity of DNA were analyzed using a spectrophotometer (NanoDrop ND-1000, Thermo Scientific, Waltham, MA, USA).

A two step-amplification approach was used to construct the 16S DNA library. The samples were amplified in 20-μL reactions, each composed of 5 μL of diluted DNA, 0.4 μM of each primer (Table 1), 0.25 mM deoxynucleotide (dNTP), 1× Phusion HF buffer, and 1 U Phusion High-Fidelity DNA polymerase (New England BioLabs, Inc., Ipswich, MA, USA). PCR was conducted in a 2720 Thermal Cycler (Applied Biosystems, Foster City, CA, USA) with 25 cycles of 95 °C for 30 s, 60 °C or 49 °C for 30 s, 72 °C for 45 s, and a final extension of 7 min at 72 °C. Three PCR replicates were performed per sample.

Products were purified using the SPRIselect Purification Kit (Beckman Coulter Life Sciences, Indianapolis, IN, USA) and, after bead purification, the target band was confirmed using 1.8% agarose gel. Barcodes were introduced by a second PCR analysis with platform-specific barcode-bearing primers. Each 50-μL PCR reaction contained the PCR product, 0.2 μM of each primer (Table 1), 0.3 mM dNTP, 1× Phusion HF buffer, and 1 U Phusion high-fidelity DNA polymerase. Ten cycles of the PCR profile described above were performed. PCR products were purified using the SPRIselect purification kit (Beckman Coulter Life Sciences, Indianapolis, IN, USA). After the final purification, each sample was quantified using a Qubit^®^ 2.0 Fluorometer (Invitrogen, Life Technologies, Monza, Italy). The amplicon library quality was tested using an Agilent 2100 Bioanalyzer (Agilent Technologies, Palo Alto, CA, USA). Libraries were sequenced using the Illumina MiSeq platform with a paired-end 300-cycle run (Macrogen Inc., Seoul, Korea).

### 3.8. Statistical and Bioinformatical Analysis

Individual data for live weight, daily growth, and slaughter results were submitted to ANOVA, with diet as a fixed effect and pen as a random effect, using the PROC MIXED procedure in SAS. Pen data for feed intake and conversion were also submitted to ANOVA, with diet, sex, and their interactions as the main factors of variability, using the PROC GLM procedure [61]. Differences between means of *p* < 0.05 were considered statistically significant.

Non-linear regressions were performed with LAB Fit Curve Fitting Software (Nonlinear Regression Program) V 7.2.50 (1999-2020) available on www.labfit.net (accessed on 5 September 2020) on [62] to evaluate sensory traits.

For data from the microbiological analysis, a multivariate statistical analysis was adopted by the non-parametric combination (NPC) test conducted with the free software NPC Test R10 (accessed on 28 March 2020) [63]. The partial and global *p*-values were estimated with diet supplementation and storage time as the main factors and time applied as a stratification block. The microbial shelf life and growth parameters were defined using primary models developed thought the Combase platform (University of Tasmania, Australia; USDA Agricultural Research Service, Washington DC, USA, https://www.combase.cc/index.php/en/, accessed on 9 December 2020) and DMfit software (https://www.combase.cc/index.php/en/, University of Tasmania, Australia; USDA Agricultural Research Service, Washington DC, USA, accessed on 9 December 2020) [64,65]. The shelf life thresholds were set at 7 log_10_ CFU/g for TVC and 7.3 log_10_ CFU/g for *Pseudomonas* (as specific spoilage organisms) [37,66].

The raw sequencing data were imported into QIIME 2 pipeline version 2019.10 (https://qiime2.org/, Northern Arizona University, Flagstaff, AZ, USA, accessed on 29 November 2020) [67] with default parameters. The 16S raw reads were denoised and dereplicated, chimera was removed, and raw reads were then merged using DADA2 software (https://benjjneb.github.io/dada2/index.html, Northern Arizona University, Flagstaff, AZ, USA, accessed on 29 November 2020). After detecting and correcting Illumina amplicon sequence errors, DADA2 was used to create a high-resolution amplicon sequence variant (ASV) table. Subsequently, the taxonomy was assigned by the scikit-learn method using a pre-trained Naive Bayes classifier against SILVA database release 132 (https://www.arb-silva.de/documentation/release-132/, accessed on 30 November 2020). Data normalization, alpha- and beta-diversity determination, and statistical analysis were conducted with CALYPSO version 8.84 (http://cgenome.net/wiki/index.php/Calypso, University of Queensland Diamantina Institute, Brisbane, QLD, Australia, accessed on 1 December 2020) [68]. The raw sequence data were deposited in the Sequence Read Archive (SRA) database (accession number PRJNA703965).

The data on drip loss and pH were analyzed by ANOVA with diet (L0, L1, L2), storage hours (24 h to 264 h), and their interaction being the main effects. When the ANOVA was significant, means were compared using the Tukey b posteriori test. Statistically significant differences were established at *p* < 0.05. The data on the proximate composition, cooking loss, FAME, and lipid oxidation were analyzed by ANOVA with diet (L0, L1, L2), meat treatment (raw and cooked), and their interaction being the main effects. The replicate effect was not significant and was removed from the model. When the ANOVA was significant, means were compared using the Tukey b posteriori test. Statistically significant differences were established at *p* < 0.05. The data were processed with IBM SPSS Statistics version 26 [69].

## 4. Conclusions

The enrichment of the diets of chickens with phenols obtained by the filtration and concentration of oil mill vegetation water for 24 days was not associated with differences in animal growth, the feed conversion rate, or the carcass yield. The study of the intestinal microbiota showed a significant effect of feeding time but not of diet per se on the alternation of microbial groups. On the other hand, the microbiota composition of the meat during the shelf-life period showed greater growth of *Pseudomonas* in the samples of chickens that consumed feed enriched with phenols. Finally, no differences were observed among diets with respect to the analytical markers (TBARs and conjugate dienes) of the lipid oxidation process. Instead, hydroxytyrosol was identified in the muscle tissue of the samples from chickens that consumed feed with added phenols, showing dose-dependent intestinal absorption. Since the oxidative process mainly occurs in the membrane structure of the muscle cells, further studies are necessary to better understand the distribution of HT in the cellular and extracellular compartments.

## Figures and Tables

**Figure 1 molecules-26-04307-f001:**
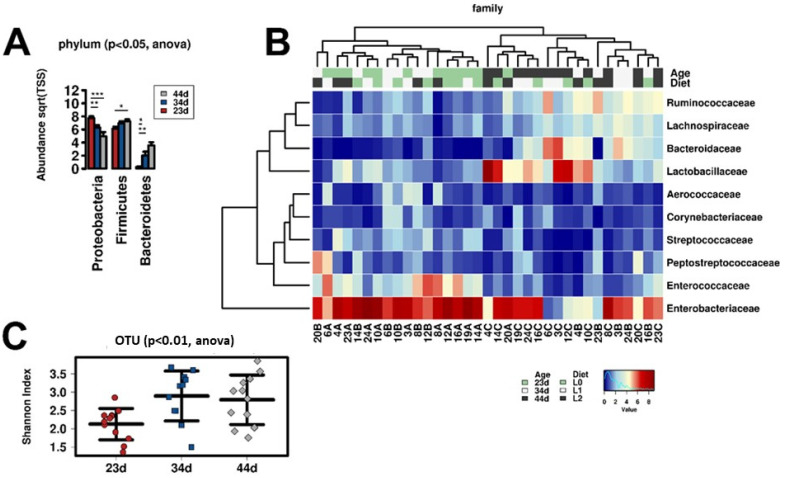
Effect of age on the gut microbiota composition. Significant variation in the relative concentrations of bacterial phyla at 23, 34 and 44 days of age (**A**), composition of the bacterial community at the family level in all cloacal swab samples (**B**), and significant variation in the biodiversity at 23, 34, and 44 days of age (**C**).

**Figure 2 molecules-26-04307-f002:**
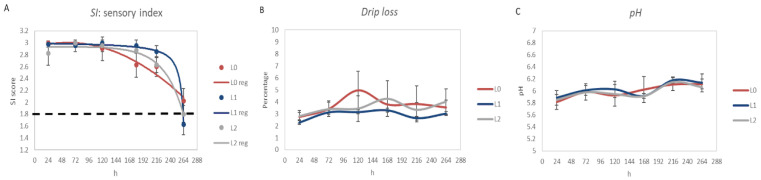
(**A**) Sensory index evaluation of raw breasts. The dotted flat line represents the shelf life threshold (1.8); (**B**) drip loss; (**C**) pH value of raw chicken breast meat.

**Figure 3 molecules-26-04307-f003:**
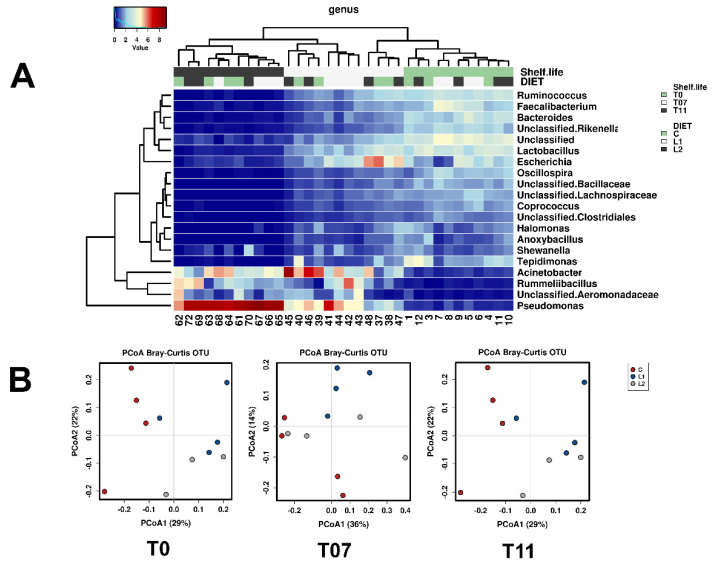
Microbial genera composition (**A**) and effect of CPC (**B**) on the microbiota composition of pectoralis major muscles of broiler chickens. In the heat map, the 20 predominant genera are reported (T0 = 24 h, T07 = 168 h; T11 = 264 h).

**Table 1 molecules-26-04307-t001:** Productive performance (least square means) of broiler chickens prior to slaughter.

	Diets			*p*-Value	SDR
	L0	L1	L2		
Broilers, n	40	40	39		
Live weight, g					
1 d	54	55	54	n.s.	4
24 d	1275	1302	1304	n.s.	32
48 d	3613	3650	3615	n.s.	230
First period (1–23 d) ^1^					
Daily weight gain, g/d	53.1	54.2	54.3	n.s.	4.5
Daily feed intake, g/d	67.1	67.1	65.6	n.s.	2.5
Feed conversion	1.27	1.23	1.22	n.s.	0.05
Second period (24–48 d) ^2^					
Daily weight gain, g/d	97.4	97.9	96.3	n.s.	9.0
Daily feed intake, g/d	193	189	190	n.s.	4.9
Feed conversion	1.98	1.94	1.98	n.s.	0.06

^1^ All animals received the same feed treatment.^2^ The three groups received different feeds: control feed without polyphenol addition (L0), L0 feed with 220 mg/kg polyphenols (L1) and L0 feed with 440 mg/kg polyphenols (L2). n.s.: *p*-value > 0.05.

**Table 2 molecules-26-04307-t002:** Yield and carcass traits in broilers slaughtered at 48 d of age.

	Diets			*p*-Value	SDR
	L0	L1	L2		
Broilers, n	40	40	39		
Slaughter live weight, g	3613	3650	3615	n.s.	230
Cold carcass, g	2856	2863	2845	n.s.	189
Dressing out percentage, %	78.8	78.2	78.5	n.s.	1.7

Control feed without phenol addition (L0), L0 feed with 220 mg/kg polyphenols (L1) and L0 feed with 440 mg/kg polyphenols (L2). n.s.: *p*-value > 0.05.

**Table 3 molecules-26-04307-t003:** Hydroxytyrosol concentration in the *P. major* muscle at the beginning (24 h) and after 11 days of the shelf-life period. Values (µg/kg) are presented as the mean ± standard deviation.

	Hours	Diet		
		L0	L1	L2
3,4-DHPEA	24	78.0 ± 5.4	185.5 ± 8.5	268.5 ± 7.7
	264	<LOD	<LOD	76.5 ± 4.0

3,4-DHPEA: Hydroxytyrosol; LOD at 278 nm 25 µg/kg.

**Table 4 molecules-26-04307-t004:** Effects of the diets (L0, L1, L2) on the estimated growth parameters (±SE) making up the total viable count and the *Pseudomonas* spp. concentration in the pectoralis major muscles of broiler chickens.

	Growth Curve Parameters				
Target	i	Lag	Log	Plateau	Estimated Shelf-Life (h)
	log_10_ (CFU/g)	H	log_10_ (CFU/g/h)	log_10_ (CFU/g)	
Total viable count **					
L0	3.9 ± 0.2	179.1 ± 12.0	0.05 ± 0.02	7.4 ± 0.3	240
L1	3.6 ± 0.3	120.7 ± 23.7	0.03 ± 0.01	7.8 *	235
L2	3.8 ± 0.2	169.6 ± 7.7	0.07 ± 0.01	7.9 ± 0.3	214
*Pseudomonas* ***					
L0	1.7 ± 0.3	-	0.02 ± 0.003	7.9 ± 3.3	264
L1	2.2 ± 0.2	71.9 ± 14.4	0.03 ± 0.003	8.3 ± 0.8	230
L2	2.2 ± 0.2	89.9 ± 14.9	0.03 ± 0.003	8.2 *	230

i = initial cell count. lag = Lag phase. log = exponential phase. Plateau = stationary phase. * estimated parameter at the end of shelf life. ** shelf-life threshold = 7 log_10_(CFU/g). *** Shelf-life threshold = 7.3 log_10_(CFU/g).

**Table 5 molecules-26-04307-t005:** Effects of diet (L0, L1, L2) and treatment (raw and cooked) on the proximate composition and cooking loss of compounds in the pectoralis major muscles of broiler chickens.

	Diet (D) ^1^			Treatment (T) ^2^		*p*-Value			
	L0	L1	L2	Raw	Cooked	D	T	D × T	SEM
Breast (n)	16	16	16	24	24				
Water (%)	72.5	72.7	72.6	74.9	70.2	n.s	<0.001	n.s	0.17
Crude protein (% FM)	24.2	24.0	23.9	21.8	26.3	n.s	<0.001	n.s	0.17
Crude fat (% FM)	1.48	1.44	1.60	1.39	1.63	n.s	<0.05	n.s	0.050
Ash (%)	1.20	1.24	1.14	1.21	1.17	n.s	n.s.	n.s	0.026
Crude protein (% DM)	88.0	87.8	87.1	87.1	88.2	n.s	n.s.	n.s	0.37
Crude fat (% DM)	5.4	5.3	5.8	5.5	5.5	n.s	n.s.	n.s	0.18
Ash (% DM)	4.2	4.6	4.2	4.8	3.8	n.s	<0.001	n.s	1.12
Cooking loss (%)	23.7	18.7	23.2			n.s			1.04

^1^ Means of the different treatments within each diet. ^2^ Means of the different diets within each treatment. L0: Control feed (without CPC); L1: L0 plus CPC (220 mg/kg total polyphenols); L2: L0 plus CPC (440 mg/kg total polyphenols). n.s.: *p* > 0.05.

**Table 6 molecules-26-04307-t006:** Effects of diet (L0, L1, L2) and treatment (raw and cooked) on the fatty acid composition (% total FA) of pectoralis major muscles of broiler chickens.

	Diet (D) ^1^			Treatment (T) ^2^		*p*-Value			
	L0	L1	L2	Raw	Cooked	D	T	D × T	SEM
Breast (n)	16	16	16	24	24				
C12:0	0.06	0.07	0.08	0.07	0.07	n.s.	n.s.	n.s.	0.006
C14:0	0.86	0.93	1.00	0.89	0.97	n.s.	n.s.	n.s.	0.033
C15:0	0.11b	0.10b	0.13a	0.09	0.14	<0.05	<0.001	n.s	0.005
C16:0	24.1	23.9	23.8	24.1	23.8	n.s	n.s	n.s	0.145
C17:0	0.20	0.20	0.20	0.20	0.20	n.s	n.s	n.s	0.004
C18:0	8.9	8.8	8.5	8.8	8.7	n.s	n.s	n.s	0.122
C24:0	1.03	0.93	0.83	0.90	0.96	n.s	n.s	n.s	0.040
Saturates	35.4	35.0	34.6	35.1	34.9	n.s	n.s	n.s	0.187
C14:1	0.18	0.17	0.18	0.17	0.18	n.s	n.s	n.s	0.004
C16:1	4.2	4.0	4.2	4.2	4.1	n.s	n.s	n.s	0.096
C17:1	0.37	0.42	0.36	0.37	0.40	n.s	n.s	n.s	0.019
C18:1	34.3	34.2	35,1	34.9	34.2	n.s	n.s	n.s	0.251
C20:1	0.07	0.08	0.09	0.07	0.09	n.s	n.s	n.s	0.005
Monounsaturates	39.2	38.9	40.0	39.7	39.0	n.s	n.s	n.s.	0.321
C18:2 n6	17.9b	19.0a	18.5ab	18.2	18.6	<0.05	n.s	n.s	0.148
C18:3 n6	0.22c	0.24b	0.27a	0.24	0.25	<0.05	n.s	n.s	0.007
C18:3 n3	1.03c	1.11b	1.21a	1.05	1.19	<0.001	<0.001	n.s	0.018
C20:2 n6	0.49	0.52	0.45	0.49	0.49	n.s	n.s	n.s	0.019
C20:3 n6	0.74	0.67	0.66	0.64	0.74	n.s	n.s	n.s	0.024
C20:3 n3	0.06	0.06	0.06	0.06	0.06	n.s	n.s	n.s	0.002
C20:4 n6	3.7	3.3	3.1	3.3	3.5	n.s	n.s	n.s	0.128
C20:5 n3	0,17	0.17	0.15	0.16	0.16	n.s.	n.s.	n.s.	0.006
C22:5 n3	0.61	0.57	0.54	0.55	0.60	n.s.	n.s.	n.s.	0.025
C22:6 n3	0.37	0.32	0.29	0.32	0.33	n.s.	n.s.	n.s.	0.015
Polyunsaturates	25.3	26.0	25.2	25.0	26.0	n.s.	n.s.	n.s.	0.277
LCPUFA	6.2	5.7	5.2	5.5	5.9	n.s.	n.s.	n.s.	0.204
n6	23.1	23.7	23.0	22.9	23.6	n.s.	n.s.	n.s.	0.242
n3	2.2	2.2	2.3	2.1	2.3	n.s.	<0.05	n.s.	0.044
n6/n3	10.5	10.8	10.3	10.8	10.2	n.s.	<0.05	n.s.	0.133

^1^ Means of the different treatments within each diet. ^2^ Means of the different diets within each treatment. L0: Control feed (without CPC); L1: L0 plus CPC (220 mg/kg total polyphenols); L2: L0 plus CPC (440 mg/kg total polyphenols). LCPUFA: sum of C20:2 n6 + C20:3 n6 + C20:3 n3 + C20:4 n6 + C20:5 n3 + C22:5 n3 + C22:6 n3 ^a,b^ Means within a row without a common superscript are different (*p* < 0.05); n.s.: *p* > 0.05.

**Table 7 molecules-26-04307-t007:** Effects of diet (L0, L1, L2) and treatment (raw and cooked) on primary (diene D232 and triene D270 conjugates) and secondary (Thiobarbituric Acid Reactive Substances, TBARs mg/kg meat) lipid oxidation compounds in the pectoralis major muscles of broiler chickens.

	Diet (D) ^1^			Treatment (T) ^2^		*p*-Value			
	L0	L1	L2	Raw	Cooked	D	T	D × T	SEM
Breast (n)	16	16	16	24	24				
TBARs	0.048	0.048	0.058	0.034	0.068	n.s.	<0.001	n.s.	0.005
D232	11.4	11.3	10.8	11.7	10.7	n.s.	<0.01	n.s.	0.178
D270	7.6	7.9	7.1	7.6	7.4	n.s.	n.s.	n.s.	0.223

^1^ Means of the different treatments within each diet. ^2^ Means of the different diets within each treatment. L0: Control feed (without CPC); L1: L0 plus CPC (220 mg/kg total polyphenols); L2: L0 plus CPC (440 mg/kg total polyphenols). n.s.: *p* > 0.05.

## Data Availability

Not applicable.

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
