# Peer review of "Employment of Phenolic Compounds from Olive Vegetation Water in Broiler Chickens: Effects on Gut Microbiota and on the Shelf Life of Breast Fillets"

_molecules, 2021, doi:10.3390/molecules26144307_

Round 1

Reviewer 1 Report

The study presents application of the olive vegetation water to supplement diet of broiler chickens Pectoralis major. Experimental design was very well thought through and executed, focusing on effects of phenolic compounds on growth performance, phenol compound composition in breast muscle, gut microbiota composition, shelf-life of fillets, sensory shelf life, pH, drip loss, proximate composition, fatty acid composition, cooking loss and fatty acid oxidation. No significant impact of diet amended by polyphenols was determined in slaughter yield, gut microbiome profile, pH, proximate composition, cooking loss, fatty acid composition (with exception of C18:2 n6, C18:3 n3), as well as in lipid oxidation. The study showed the effect of age on the gut microbiota profile and the effect of CPC on microbial composition during storage. The L1 and L2 diets were associated with a higher concentration of Pseudomonas and sensory decay. The study relies on up-to-date analytical approaches, microbial and sensory analysis, sequencing, etc. The authors declare adherence to ethical principles and treatment of animals.

The manuscript is well written but some less clear sentence structure or typos can be found:

51 …..presents …..

58 ….In virgin olive oil,  …..

69 …..presents…..

76……, that might operate in a reciprocal …..

96 …….are is available

156 …..seemed to not to affect….

167 ……digesta (?) transit….

224 ….the effects of the inclusion of phenols….

237…..microbial growth shape curve….

255-257  The sentence is not very clear.

273-274  ….evidenced a significant increase significantly increased protein content and decreased fat….

Frequent typos:

  1. Pseudomonas but Pseudomonas

Author Response

We are grateful for the suggestions to improve the readability of the manuscript. All the indications have been accepted, typos included, and the manuscript has been updated accordingly, using the color green to distinguish the updates.

Reviewer 2 Report

The article “Employment of phenolic compounds from olive vegetation water in broiler chickens: effects on gut microbiota and on the shelf life of breast fillets” submitted to “Molecules” provides very detailed insights into the effect of feeding broiler chickens with a phenolic concentrate from olive vegetation water. Besides growth, fat content, chemical composition and shelf live, also gut microbial composition was determined via sequencing. The applied methods were clearly described and the experimental procedure was well thought through. The graphic representation is appealing and all relevant data are presented in tables and in the supplements. I only found three minor points, which could be addressed prior publication:

  • Line 58-60: some words and/or punctuation marks are missing
  • Line 175-180: Please explain what the demerit threshold is
  • General: I miss a broader discussion about the meaning of your experiments for human consumers. Olive polyphenols are known for their beneficial health effects, for instance regarding Parkinson’s and Alzheimer’s disease. Do you think, the polyphenol concentration found in the tested chicken meat would be sufficient to produce positive effects in humans after consumption? A short discussion about it would be desirable.

Author Response

The authors' responses are in the attached file
